# Transformers Learn Transition Dynamics when Trained to Predict Markov Decision Processes

**Yuxi Chen*** and **Suwei Ma*** and **Tony Dear**
Department of Computer Science
Columbia University
New York, NY 10027
{yc4041, sm5011, tbd2115}@columbia.edu

**Xu Chen**
Department of Electronic Engineering
Tsinghua University
Beijing, China 10084
chenxu323@tsinghua.edu.cn

## Abstract

Language models have displayed a wide array of capabilities, but the reason for their performance remains a topic of heated debate and investigation. Do these models simply recite the observed training data, or are they able to abstract away surface statistics and learn the underlying processes from which the data was generated? To investigate this question, we explore the capabilities of a GPT model in the context of Markov Decision Processes (MDPs), where the underlying transition dynamics and policies are not directly observed. The model is trained to predict the next state or action without any initial knowledge of the MDPs or the players' policies. Despite this, we present evidence that the model develops emergent representations of the underlying parameters governing the MDPs.[1]

## 1 Introduction

Recently, large language models (LLMs) have gained significant popularity and attention due to their versatility and performance, including in writing code, engaging in meaningful conversations, and much more. Many of these models, trained on the simple principle of "predicting the next word," go on to become vastly capable polymaths. Yet the reason behind how language models come to obtain this performance remains a subject of continuous debate and research.

Many have suggested, based on the extensive number of parameters of these language models, that their performance may result from merely memorizing "surface statistics," or external correlations that do not necessarily reflect the underlying data generation process. Such issues can arise, for instance, when the pre-training corpora contains frequently co-occurring words, which can be preferred over the right answer (Kang and Choi, 2023).

Another instance in which a language model has been shown to learn causal statistical dependencies is due to dataset selection bias (McMilin, 2022).

It has also been suggested that language models can construct world models—interpretable and internal characterizations of the environment from which the data generating process is derived (Goldstein and Levinstein, 2024). Recent works have shown that LLMs are able to develop internal representations of concepts such as color (Abdou et al., 2021) and direction (Patel and Pavlick, 2021).

A standard way to evaluate the emergence of internal representations of the world state in these models is to assess them in a relatively well-behaved, self-contained environment in which the rules are clearly stated and understood. To illustrate, Toshniwal et al. (2021) have explored how such models, trained on sequences of chess moves, are able to predict valid chess moves with high accuracy. The authors also suggest that the model keeps track of the current board state for the prediction step. Li et al. (2022) extended this idea by exploring the internal representations of a GPT-2 variant trained on the game of Othello.

However, previous works have only investigated how these models are able to internally identify the current state and stop short of demonstrating whether they are able to identify parameters governing the underlying data generation process. *The main goal of this paper is to take a step towards filling in this gap in the context of Markov Decision Processes (MDPs), where the sequence of states and actions are generated by hidden, parameterized policies and transition dynamics.*

Specifically, we consider the synthetic and well-understood game of ConnectFour for our investigation. First, we generate data in the form of game transcripts where the both players follow a policy guided by either Deep $Q$-Learning (DQL) or Monte-Carlo Tree Search (MCTS). Then, we train 3 transformer models each when the game

---

*equal contribution
[1] https://github.com/YuxiChen25/TF-MDP

transcript is represented using only the states (co-ordinates of the played pieces) or actions (which column the piece is placed in), hence totaling 12 transformers.

Next, we investigate whether the transformer models trained on the game transcripts contain an internal representation of the parameters governing the transition dynamics, which takes the form of either the players' deep $Q$-Values or MCTS values. We verify whether the model is able to identify a salient representation when predicting the next state or action conditioned on the partial transcript seen thus far via probing—training classifiers to predict the deep $Q$-values or MCTS values of the current game state using the network's internal activations as input (Alain and Bengio, 2016; Tenney et al., 2019). Using this probing technique, we find ample evidence of these models being able to internally represent the generative process despite changing the transition dynamics and representation of the input data.

In summary, our contributions are twofold: 1) we show evidence that transformer models contain internal representations of the underlying transition dynamics governing Markov Decision Processes after trained to predict the next tokenized state or action 2) we show that our result is robust to how this process is represented to the transformer model as input data and how the policy of the MDP is defined.

## 2 Dataset Generation and Language Modeling

We focus on investigating internal representations of language models in a well-understood, self-contained synthetic game setting. This is motivated by the observation from past works that the language models learn to predict valid game moves by simply being trained to extend game transcripts (Toshniwal et al., 2021; Li et al., 2022). Specifically, we select ConnectFour, a turn-based, two-player, board-completion game in which the goal is to connect four pieces of a player's own color. The ConnectFour environment is shown in Figure 1.

In ConnectFour, the game is played on a $6 \times 7$ board where two agents place alternating pieces of red or yellow discs on the board, which fall down to the bottom-most unoccupied row of the column chosen by the agent. The objective for both agents is to connect four pieces of the same color before the opponent, whether horizontally, vertically, or

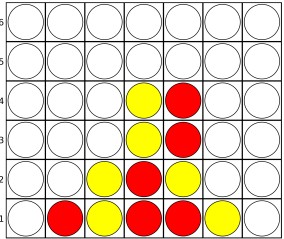

Figure 1: ConnectFour

diagonally. The agent who wins the game secures a terminal reward.

We choose this environment for two reasons: first, ConnectFour has a game tree that is exponentially large, hence making it infeasible for any transformer model to brute-force "memorize" or "recite" the optimal game-play strategy for all outcomes; second, training deep reinforcement learning agents or growing Monte-Carlo search trees on ConnectFour to approximate optimal playing strategies have been shown to enjoy good performance (Alderton et al., 2019; Sheoran et al., 2022).

### 2.1 Generation of Game Transcripts

We describe below how game transcripts are generated using when players are trained on deep $Q$-learning or guided by MCTS in the ConnectFour environment to be used to autoregressively train our transformer models. Then, we give a high-level overview of MDPs and its connection to our setting.

#### 2.1.1 Deep $Q$-Learning

To start, we use deep $Q$-learning (DQL) to train both players in the ConnectFour environment since traditional $Q$-learning often struggles to converge in when the space of outcomes is combinatorially large (Mnih et al., 2013). We define a neural network parameterized by weights $\theta$, which takes the current state $s$, and action $a$ and outputs a scalar value $Q_\theta(s, a)$. The state space $\mathcal{S}$ consists of all possible configurations of the $6 \times 7$ board, while the action space $\mathcal{A}$ is defined as placing a disc in the $i$-th column, where $i \in \{1, 2, \ldots, 7\}$.

Our training process is based on a variant of the original deep $Q$-learning algorithm. Specifically, the architecture of our network $Q_\theta$ consists of one convolutional layer followed by two linear layers. The network predicts the $Q$-values for placing a disc in each of the seven columns. See algorithm 1 in Appendix A for further training details. We train nine pairs of RL agents, each pair competing

against each other for one million games. For each game, we record the action (the column into which the piece is placed), the state (represented henceforth as coordinate of the played piece), and the deep $Q$-values of all the feasible moves at each step (if a move is infeasible, then the value is set to 0). Then, we combine the last 111K games played for every pair totaling one million game transcripts.

### 2.1.2 Monte-Carlo Tree Search

We also generate data using MCTS. MCTS is a heuristic search algorithm that has shown remarkable success in classic board-games, modern-board games, and video games. MCTS combines depth-first search and stochastic simulation to build and use a game tree of possible outcomes based on selection, expansion, simulation, and back-propagation (Chaslot et al., 2008). In our setting, we implement the standard MCTS algorithm where at each move decision, we run 100 rollouts, and the action with the highest MCTS value (win rate) is selected. Similar to the above, we generate one million game transcripts by running 1 million independent ConnectFour games where both agents play according to the MCTS heuristic. We also record the action, state, and corresponding MCTS values at each step (the value is likewise 0 if the move is infeasible).

### 2.2 Connection to MDPs

A Markov Decision Process (Ghavamzadeh et al., 2015) $\mathcal{M}$ is a tuple $\langle \mathcal{S}, \mathcal{A}, P, P_0, R \rangle$ where $\mathcal{S}$ is the set of states, $\mathcal{A}$ is the set of actions, $P(\cdot|s, a) \in \mathcal{P}(\mathcal{S})$ is the probability distribution over next states, conditioned on action $a$ being taken in state $s$, $P_0 \in \mathcal{P}(\mathcal{S})$ is the probability distribution according to which the initial state is selected, and $R(s, a) \in \mathcal{P}(\mathbb{R})$ is a random variable representing the reward obtained when action $a$ is taken in state $s$. A policy—a mapping from past observations to a distribution over the set of actions—is a rule for choosing actions at any given state. Policies can be characterized as

1. *Markov* if the distribution is only dependent on the last state of the observation sequence.

2. *Stationary* if the distribution does not change over time.

3. *Deterministic* if the probability distribution concentrates on a single action for all possible histories of states and actions.

A Markov Decision Process is called *first-order* if the state transition probability $P(s_{t+1}|H_t = s_1, a_1, \ldots, s_t, a_t) = P(s_{t+1}|s_t, a_t)$ depends only on the latest state and action and *n-th* order if $P(s_{t+1}|H_t) = P(s_{t+1}|s_1, a_1, \ldots, s_{t-n}, a_{t-n})$ depends on the last $n$ states and actions.

In the ConnectFour setting, we can define the components of the MDP as

- $\mathcal{S}$: All possible board configurations.

- $\mathcal{A}$: Valid column placements out of the 7 columns on the ConnectFour board.

- $P$: Deterministic transitions based on player actions.

- $P_0$: The initial empty ConnectFour board state.

- $R$: Reward based on game outcome, which can either be a win, loss, or draw.

Having outlined the above, we see that players guided by deep $Q$-learning in ConnectFour follow a policy that is

- *Markov*: The neural network only considers the current board state as input.

- *Non-stationary*: The network's parameters are continuously updated during training, which means that their game-playing strategy can also evolve.

- *Non-deterministic*: This is due to $\epsilon$-greedy exploration, mini-batch sampling, and other sources of randomness during training.[2]

The resulting MDP is first-order, as the next state depends only on the current state and action.

Players guided by MCTS, on the other hand, can be viewed in two ways. If we consider the Monte-Carlo Search Tree as part of the state, then the policy is

- *Markov*: The current game tree and board state completely determine the distribution over the next action.

- *Non-stationary*: The search tree is able to grow over time and output different moves.

- *Non-deterministic*: This is due to the inherent stochastic nature of MCTS simulations.

---

[2]However, if all these random factors are controlled, the policy becomes deterministic.

In this worldview, the MDP under MCTS remains first-order. However, if we regard the search tree as external to the state, then the policy and MDP become $n$-th order. This is because the search tree's evolution stochastically depends on all previous moves and simulations.

In ConnectFour, given a chosen action, the next board state and terminal reward is deterministic. Therefore, the stochasticity in the MDP formulations of both deep $Q$-learning and MCTS are attributed only to randomness in the policy parameters. This means that a transformer model that internally characterizes deep $Q$-values or MCTS values with accuracy effectively captures the transition dynamics $P(\cdot|s,a)$. From here, we can conclude that such a model has a internal representation of the underlying parameters governing the MDP that generates the observed data. This insight guides our later experiments.

### 2.3 Language Modeling and Training

For both settings, each state and action are tokenized as input. We supply no further auxiliary information during training, as our goal is to study how much they can infer the underlying transition dynamics from only information of the observed histories. Each history is treated as a sentence tokenized with a predefined vocabulary (for states, this corresponds to 42 possible coordinates of the discs; for actions, this corresponds to the 7 column placements; an extra padding vocabulary is included for both).

For each setting (deep $Q$-Learning and MCTS) and each representation of the history (using state or action), we train three separate 8-layer GPT models (Radford et al., 2018) with an 8-head attention mechanism and a 512-dimensional hidden space. When we represent the history using only actions, we let the transformer predict state $s_t$ conditioned on the history $\{s_1, \ldots, s_{t-1}\}$. In the action-exclusive setting, we let the transformer predict action $a_t$ conditioned on $\{a_1, \ldots, a_{t-1}\}$.[3] The models' weights are initialized randomly, including the layer for word embeddings.

Training is next performed autoregressively: for each tokenized partial history where each element is either a state or action, the forward process converts the input via the trainable word embedding

---

[3] As mentioned above, we want to explore both representations to see if the transformers' model's learning of the parameters of the MDP, if successful, is robust to how the input data is represented during training.

into $\{c_t^0\}_{t=1}^{T-1}$, where $c_t^i$ is the intermediate feature for the $t$-th token after the $i$-th layer to be sequentially processed by 8 multi-head attention layers. Using a causal mask, we ensure that only $c_{\leq t}^{i-1}$ are visible to $c_t^i$ during training; that is, the prediction step only involves features in the preceding layer and earlier time steps. $c_{T-1}^8$ is lastly fed through a linear layer to predict logits for the ground-truth state or action. We use cross-entropy loss between the predicted logits and the ground-truth state or action as the objective during training. The parameters of the network are optimized by gradient descent, and we use the model weights corresponding to the epoch with the lowest validation loss to explore internal representations.[4]

### 3 Exploring Internal Representations

As mentioned above, to see if our language model effectively captures the underlying transition dynamics of the Markov Decision Processes, we use a standard tool called "probing," which is the process of training a classifier or regressor using the internal activations of a transformer model as input features to predict labels or values of interest. If we can train probes in all four settings (whether the policy is governed by MCTS or deep $Q$-learning and whether the MDP is represented using states or actions), then we can conclude that the transformer models effectively internally characterize information about the parameters governing the MDPs.

### 3.1 Experimental Setup

To train all probes, we first randomly sample one time stamp $t$ in each game to obtain partial histories $\mathcal{H}_{t-1} = \{s_1, \ldots, s_{t-1}\}$ or $\mathcal{H}_{t-1} = \{a_1, \ldots, a_{t-1}\}$. Then, we retrieve the corresponding internal embedding $E_i^t$ of the network that is used to predict $s_t$ or $a_t$ when the input is $\mathcal{H}_{t-1}$ after the $i$-th layer of the network. We repeat this process of retrieving the embedding after every layer of the network, and obtain $\{E_i^t\}_{i=1}^8$ for each sampled partial history $\mathcal{H}_{t-1}$. We repeat this process 679K times for each probe and split the dataset into training, validation, and testing data according to a 8-1-1 split. The embeddings after each layer are used to train separate probes, that is, we use $\{E_1^t\}_t$ to train the probe who uses embedding information output by the transformer after the first layer, $\{E_2^t\}_t$ to train the probe that uses the em-

---

[4] See Appendix B for more training details.

beddings after the second layer, totaling 8 probes for any particular combination of policy and data representation. We also repeat the probe training process 3 times for any setting corresponding to the 3 transformer models trained in each setting.

To train the probes, we use the embedding $E_i^t$ as input to regress against the true corresponding deep $Q$-values or MCTS values underlying the MDP at the time-step $t-1$ given the partial history $\mathcal{H}_{t-1}$. For example, suppose at time $t-1$ that a player's MCTS values used to make the decision at time $t$ are $m_{t-1} = (0.2, 0.4, 0.7, 0.9, 0.1, 0.1, 0.1)$ corresponding to columns 1 through 7. The player would have chosen action $a_t = 4$ or $s_t = (4, 2)$[5] since the MCTS value corresponding to column 4 is highest. Then, we extract the embedding $E_i^t$ associated with predicting $a_t$ or $s_t$ and use it to regress the 7-dimensional vector $m_{t-1}$. The parameters of each probe is optimized by gradient descent, and we select the model weights with the lowest validation loss to explore our hypothesis.

Inspired by Li et al. (2022), we also explore if the performances of linear and non-linear probes have significantly different accuracies, which may suggest how the parameters of the MDP are represented in the transformer model. In both settings, we compare probe performance trained on internal activations after each layer against a probe trained and validated on randomly generated embeddings.[6] It is clear that probes trained even on randomized embeddings may perform better than blindly "guessing" a random real-valued vector.[7] This approach allows us to see whether a random probe can encode information about the parameters of the MDP without any additional data as good as a properly trained probe. If the test loss between the two types of probes are indistinguishable, then this suggest that the transformers' internal activations do not contain any effective information of the MDP parameters.

For linear probes, the prediction of the deep $Q$-values or MCTS values parameterized by weights $\theta$ is given by $W E_i^t$ where $\theta = \{W \in \mathbb{R}^{D \times d}\}$, $D =$

---

[5]Here, we suppose there already exists a disc beneath it played before, hence the current $y$-coordinate is 2.

[6]Each entry in the embedding is drawn independently from a normal distribution with mean of 0 and standard deviation of 5. We refer to these probes as "random probes" hereinafter for concision.

[7]Since even a network with random valued vectors as input can encode the empirical mean of the observed data. Then, if the distribution of the training and testing data are the same, we should expect to see that this network performs better than blindly guessing.

512 is the number of dimensions of the internal embedding $E_i^t$ and $d = 7$ is the dimension of the output space. For nonlinear probes, the prediction can be written as $W_1 ReLU(W_2 E_i^t)$, where $\theta = \{W_1 \in \mathbb{R}^{D \times d}, W_2 \in \mathbb{R}^{D \times D}\}$.

### 3.2 Empirical Evaluations

We verify the performance of our probes on 2 different metrics: 1) mean squared error between the predicted and ground truth deep $Q$-values or MCTS values of the moves 2) whether the best move predicted by a probe matches the ground-truth best move. We do not normalize any of the ground-truth or predicted values prior to evaluation.

#### 3.2.1 Mean Squared Error

We first show the test MSE loss when the trained probes regress against the target deep $Q$-values or MCTS values of the player. The first column denotes the layer of the transformer model after which the embeddings are used to train the probe, where "R" stands for randomly generated embeddings. We report the mean and standard deviation of the test losses obtained from the three probes in each scenario in Tables 1-4. We see that trained probes have a significantly lower test loss compared to random probes across all settings, which strongly suggests that the internal activations do contain representations of the MDP parameters. We also see that non-linear probes consistently yield lower losses than linear ones, which suggests that the MDP parameters may admit a non-linear representation in the transformer models. In addition, the difference in the scale between the DQL and MCTS settings can be easily explained: while MCTS values are bounded between 0 and 1, it is known that conventional Deep $Q$-learning is impacted by an overestimation bias (Hessel et al., 2017). Nevertheless, our conclusion remains valid since all the probes in each setting are trained to regress against values generated from the same space.

In terms of robustness to data representation, we see how the losses of the probes when the data is being represented using only states or using only actions do not differ significantly: the non-linear layers when data is represented using states performs slightly better than that using actions. This intuitively makes sense since encoding using states inherently provide more explicitly information (since they include the $y$-coordinate of the played discs) compared to actions.

Table 1: MSE | DQL | $\mathcal{H}_{t-1} = \{s_1, \ldots, s_{t-1}\}$

|   | Linear | Non-Linear |
|---|--------|------------|
| 1 | $528.6 \pm 0.02$ | $514.0 \pm 0.06$ |
| 2 | $494.1 \pm 1.49$ | $362.2 \pm 1.80$ |
| 3 | $477.5 \pm 1.27$ | $338.4 \pm 2.28$ |
| 4 | $469.5 \pm 1.01$ | $328.5 \pm 0.95$ |
| 5 | $467.2 \pm 2.71$ | $327.2 \pm 1.00$ |
| 6 | $466.1 \pm 0.31$ | $327.9 \pm 1.66$ |
| 7 | $466.3 \pm 0.95$ | $328.6 \pm 1.83$ |
| 8 | $467.8 \pm 0.89$ | $328.8 \pm 1.95$ |
| R | $1306.6 \pm 0.06$ | $1224.5 \pm 0.41$ |

Table 2: MSE | DQL | $\mathcal{H}_{t-1} = \{a_1, \ldots, a_{t-1}\}$

|   | Linear | Non-Linear |
|---|--------|------------|
| 1 | $496.2 \pm 0.02$ | $493.5 \pm 0.02$ |
| 2 | $475.2 \pm 1.41$ | $395.8 \pm 1.12$ |
| 3 | $465.1 \pm 2.63$ | $357.3 \pm 2.83$ |
| 4 | $462.5 \pm 1.88$ | $340.8 \pm 1.74$ |
| 5 | $462.3 \pm 1.44$ | $343.0 \pm 2.70$ |
| 6 | $461.9 \pm 1.31$ | $346.1 \pm 4.38$ |
| 7 | $461.4 \pm 1.23$ | $345.9 \pm 3.88$ |
| 8 | $462.4 \pm 1.01$ | $348.8 \pm 3.02$ |
| R | $1306.6 \pm 0.06$ | $1224.5 \pm 0.41$ |

Table 3: MSE | MCTS | $\mathcal{H}_{t-1} = \{s_1, \ldots, s_{t-1}\}$

|   | Linear | Non-Linear |
|---|--------|------------|
| 1 | $0.0419 \pm 1.0$ e-7 | $0.0411 \pm 4.0$ e-6 |
| 2 | $0.0323 \pm 1.7$ e-4 | $0.0206 \pm 2.1$ e-5 |
| 3 | $0.0290 \pm 1.9$ e-4 | $0.0196 \pm 5.6$ e-5 |
| 4 | $0.0273 \pm 1.6$ e-4 | $0.0191 \pm 8.8$ e-5 |
| 5 | $0.0270 \pm 1.6$ e-4 | $0.0191 \pm 1.0$ e-4 |
| 6 | $0.0269 \pm 1.6$ e-4 | $0.0189 \pm 4.7$ e-5 |
| 7 | $0.0270 \pm 1.2$ e-4 | $0.0190 \pm 5.9$ e-5 |
| 8 | $0.0272 \pm 1.3$ e-4 | $0.0191 \pm 3.7$ e-5 |
| R | $1.4103 \pm 1.0$ e-7 | $1.4259 \pm 1.1$ e-4 |

Table 4: MSE | MCTS | $\mathcal{H}_{t-1} = \{a_1, \ldots, a_{t-1}\}$

|   | Linear | Non-Linear |
|---|--------|------------|
| 1 | $0.0420 \pm 3.0$ e-6 | $0.0416 \pm 1.0$ e-6 |
| 2 | $0.0341 \pm 7.6$ e-5 | $0.0269 \pm 7.9$ e-5 |
| 3 | $0.0309 \pm 2.0$ e-4 | $0.0224 \pm 2.7$ e-4 |
| 4 | $0.0297 \pm 1.7$ e-4 | $0.0205 \pm 3.8$ e-5 |
| 5 | $0.0285 \pm 1.1$ e-4 | $0.0202 \pm 1.3$ e-4 |
| 6 | $0.0275 \pm 1.8$ e-4 | $0.0196 \pm 1.0$ e-4 |
| 7 | $0.0267 \pm 2.3$ e-4 | $0.0195 \pm 8.2$ e-5 |
| 8 | $0.0261 \pm 2.3$ e-4 | $0.0195 \pm 9.3$ e-5 |
| R | $1.4103 \pm 1.0$ e-7 | $1.4259 \pm 1.1$ e-4 |

### 3.2.2 Correctly Identifying the Best Move

Here, we would like to investigate whether the best move predicted by the probe matches the best ground-truth move. We define the loss function as

$$\mathbf{1}[\text{Best Predicted Move} \neq \text{True Best Move}]$$

In other words, we want to see whether

$$\arg\max_i \tilde{v}_i \neq \arg\max_i v_i$$

for $i \in \{1, 2, \ldots, 7\}$ where $\tilde{v}, v \in \mathbb{R}^7$ are our predicted and ground-truth target deep $Q$-values or MCTS values respectively. We report the mean and standard deviation of the test losses across different settings in Tables 5-8. Here, we observe that the performance of the trained probes significantly excel that of the random probes, meaning that the embeddings also contain internal information on how to make the best moves.[8]

In terms of robustness to data representation, we see how the data encoded using only actions yield a lower loss compared to that of states. This may be explained by how encoding the input data using actions is more directed towards identifying the best move, since the dimensionality of the space of actions and the space of best moves are identical and their structure may hence share greater similarity. Nevertheless, both ways of representing the input data to the transformer exceeds the performance of random probes.

### 3.3 Alternative Loss Functions

It should also be noted that mean-squared error and correctly identifying the best move are not

---

[8]In addition, our probes are trained to minimize the MSE between the predicted and target values, not cross-entropy loss of the predicted and actual best move. This also implies that minimizing MSE can help partially achieve this functionality.

Table 5: BEST | DQL | $\mathcal{H}_{t-1} = \{s_1, \ldots, s_{t-1}\}$

|   | Linear | Non-Linear |
|---|--------|------------|
| 1 | $0.3892 \pm 7.4$ e-5 | $0.3494 \pm 2.0$ e-3 |
| 2 | $0.4070 \pm 9.6$ e-3 | $0.4398 \pm 5.3$ e-3 |
| 3 | $0.4419 \pm 1.1$ e-2 | $0.4620 \pm 4.4$ e-3 |
| 4 | $0.4646 \pm 1.5$ e-2 | $0.4486 \pm 1.8$ e-2 |
| 5 | $0.4723 \pm 5.8$ e-3 | $0.4623 \pm 1.4$ e-2 |
| 6 | $0.4720 \pm 4.3$ e-3 | $0.4599 \pm 2.8$ e-3 |
| 7 | $0.4751 \pm 7.0$ e-3 | $0.4487 \pm 5.0$ e-3 |
| 8 | $0.4739 \pm 5.2$ e-3 | $0.4548 \pm 7.2$ e-3 |
| R | $0.8264 \pm 5.1$ e-3 | $0.5698 \pm 6.0$ e-2 |

Table 6: BEST | DQL | $\mathcal{H}_{t-1} = \{a_1, \ldots, a_{t-1}\}$

|   | Linear | Non-Linear |
|---|--------|------------|
| 1 | $0.3489 \pm 4.5$ e-4 | $0.3013 \pm 2.2$ e-3 |
| 2 | $0.3785 \pm 1.3$ e-2 | $0.3682 \pm 2.2$ e-3 |
| 3 | $0.3887 \pm 6.1$ e-3 | $0.3803 \pm 7.7$ e-3 |
| 4 | $0.4176 \pm 2.1$ e-3 | $0.4020 \pm 1.0$ e-2 |
| 5 | $0.4283 \pm 9.5$ e-3 | $0.4034 \pm 1.0$ e-2 |
| 6 | $0.4324 \pm 7.4$ e-3 | $0.4016 \pm 1.5$ e-2 |
| 7 | $0.4305 \pm 8.0$ e-3 | $0.3936 \pm 6.4$ e-3 |
| 8 | $0.4335 \pm 2.9$ e-3 | $0.4043 \pm 1.1$ e-2 |
| R | $0.8264 \pm 5.1$ e-3 | $0.5698 \pm 6.0$ e-2 |

Table 7: BEST | MCTS | $\mathcal{H}_{t-1} = \{s_1, \ldots, s_{t-1}\}$

|   | Linear | Non-Linear |
|---|--------|------------|
| 1 | $0.0346 \pm 1.0$ e-7 | $0.0346 \pm 1.0$ e-7 |
| 2 | $0.0733 \pm 9.2$ e-4 | $0.0366 \pm 4.5$ e-4 |
| 3 | $0.0784 \pm 1.6$ e-3 | $0.0388 \pm 5.8$ e-4 |
| 4 | $0.0789 \pm 1.7$ e-4 | $0.0406 \pm 2.2$ e-4 |
| 5 | $0.0847 \pm 1.3$ e-3 | $0.0410 \pm 4.0$ e-4 |
| 6 | $0.0894 \pm 2.1$ e-3 | $0.0424 \pm 4.6$ e-4 |
| 7 | $0.0929 \pm 2.6$ e-3 | $0.0431 \pm 1.0$ e-4 |
| 8 | $0.0959 \pm 1.9$ e-3 | $0.0437 \pm 7.4$ e-4 |
| R | $0.8180 \pm 4.6$ e-3 | $0.8284 \pm 9.8$ e-4 |

Table 8: BEST | MCTS | $\mathcal{H}_{t-1} = \{a_1, \ldots, a_{t-1}\}$

|   | Linear | Non-Linear |
|---|--------|------------|
| 1 | $0.0346 \pm 1.0$ e-7 | $0.0346 \pm 1.0$ e-7 |
| 2 | $0.0347 \pm 7.5$ e-5 | $0.0360 \pm 3.6$ e-4 |
| 3 | $0.0355 \pm 2.6$ e-4 | $0.0361 \pm 1.6$ e-4 |
| 4 | $0.0375 \pm 5.2$ e-4 | $0.0362 \pm 9.6$ e-5 |
| 5 | $0.0459 \pm 2.1$ e-3 | $0.0367 \pm 5.9$ e-4 |
| 6 | $0.0558 \pm 3.5$ e-3 | $0.0376 \pm 6.7$ e-4 |
| 7 | $0.0636 \pm 3.6$ e-3 | $0.0382 \pm 3.2$ e-4 |
| 8 | $0.0678 \pm 1.6$ e-3 | $0.0386 \pm 3.0$ e-4 |
| R | $0.8180 \pm 4.6$ e-3 | $0.8284 \pm 9.8$ e-4 |

necessarily the optimal loss functions to evaluate the extent to which the model has captured the structural properties of the transition dynamics. As an illustrative example, consider when the ground truth values are

$$v = (0.9, 0.7, 0.2, 0.3, 0, 0.5, 0.4)$$

in addition to two candidate predictions

$$\tilde{v} = (0.7, 0.8, 0.2, 0.35, 0, 0.5, 0.4)$$

$$\hat{v} = (0.8, 0.6, 0.05, 0.4, 0, 0.1, 0.2)$$

We see that while the first prediction $\tilde{v}$ fails to capture the best move, it learns to predict the values of other moves with little-to-no error. In contrast, the second prediction $\hat{v}$ identifies the best move, but learns the other moves with much less precision. However, it is often unclear which of these predictions can be considered better, since they surpass the other under a different evaluation metric.

To address this issue, one potentially appealing alternative to consider may be the *Rank-Biased Overlap* (Webber et al., 2010). We define $\phi_i := (j : -v_{(j)} = -v_i)$ to be the rank of the $i$-th feature, with 1 being the best (since higher MCTS or deep $Q$-values correspond to more promising moves) and 7 being the worst. Then, we define $\tau_i := (j : \phi_j = i)$ to be the feature corresponding to rank $i$. The predicted ranks and corresponding features $\hat{\phi}, \hat{\tau}$ are defined similarly. Then, given a parameter $0 < p < 1$, the rank-biased overlap is given by

$$RBO(\{\hat{\tau}_i\}_{i=1}^7, \{\tau_i\}_{i=1}^7)$$
$$:= (1-p) \sum_{s=1}^7 p^{s-1} \frac{|\{\hat{\tau}_i\}_{i=1}^s \cap \{\tau_i\}_{i=1}^s|}{s}$$

The output is bounded between 0 and 1 and captures how well the values of our predicted moves

match those of the ground truth with regards to their orderings. It is clear that this metric, while not evaluating the numerical differences at each index, is able to preserve some notion of structural similarity between the predicted and ground-truth values. By varying $p$ from close to 0 to close to 1, one is able to interpolate between putting emphasis on only the best move to virtually all the moves.

We also remark that the choice of the evaluation metric may be highly problem-specific. For instance, one may resort to evaluation using the Kullback-Leibler (KL) divergence when the outputs are or can be normalized to probability distributions. We defer investigating alternative choices of metrics and their properties for future works.

## 4   Conclusion

In summary, our study provides compelling evidence that transformer-based models, when trained on data generated from a Markov Decision Processes, are able to develop internal representations of the underlying parameters governing these processes. Our investigation, primarily focused on the game of ConnectFour, shows that these models are able to capture information about the players' policies and hence transition dynamics of the MDPs, whether they are guided by deep $Q$-learning or Monte Carlo Tree Search, and is robust to how the data is being fed as input to the transformer model.

Specifically, we show that 1) probes trained on the internal activations of our transformer models always outperform random probes in predicting the deep $Q$-values or MCTS values, which suggests that the model encode meaningful information about the MDP parameters 2) the superior performance of non-linear probes suggest that the internal representation of MDP dynamics may have a non-linear structure within the transformer models 3) the probes' ability to identify the best move using the embeddings further support this hypothesis that they capture salient features of the MDPs 4) the robustness of these findings across different input representations and types of policy underscores the generality of our result.

We hope this study contributes to the ongoing debate about the capabilities of language models, providing evidence that they can develop rich internal representations of underlying data-generating processes. As technological advancements continue to push the boundaries of what these models can achieve, understanding their internal mecha-

nisms becomes increasingly crucial. We also wish to extend this work in the future to where there is even greater variability within the generative process, and consider alternative evaluation metrics to provide more insight along this line of research.

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

## A  Deep $Q$-Learning Algorithm

As mentioned above, to train our players using deep $Q$-Learning, we use Algorithm 1 shown below. The DQN architecture consists of a single convolutional layer followed by two fully connected layers and an output layer, designed to map 2D input states to Q-values for 7 possible actions. In our implementation of the network, we use a replay buffer of size 1000, and a mini-batch size of 32. We select actions using an epsilon-greedy policy, with $\epsilon = 0.1$.

## B  Language Modeling Details

### B.1  Dataset and Data Representations

We trained our transformers models on datasets consisting of 1 million games, with each game represented in four different forms: (1) sequences of states generated by deep $Q$-Learning, (2) sequences of states generated by MCTS, (3) sequences of actions generated by deep $Q$-Learning, and (4) sequences of actions generated by MCTS. Every dataset is split into 80% for training, 10% for validation, and 10% for testing.

### B.2  Model Architecture

The transformer model we use have a block size of 42, an embedding dimension of 512, and 8 attention heads for a total of 8 layers with a predefined vocabulary size. The dropout rates are 0.1 for embedding dropout, 0.1 for residual dropout, and 0.1 for attention dropout. The model consists of an embedding layer, followed by a series of transformer blocks, each containing a causal self-attention mechanism and a feed-forward neural network. The final layers include layer normalization and a linear projection to the vocabulary size.

### B.3  Training Procedure

For each of the four dataset representations, three transformers with identical architectures were trained to account for variability and potential error. This resulted in a total of 12 trained transformers. The models were optimized using Adam with a learning rate of 0.001, and training was conducted for 15 epochs with a batch size of 32. The loss function used is cross-entropy, calculated between the predicted logits and the true next tokens in the sequence.

### B.4  Computational Resources

All training was performed on instances equipped with 8 NVIDIA RTX 4090 GPUs.

**Algorithm 1** Training ConnectFour with Deep $Q$-Learning
___
**Input:** Number of episodes $M$, number of game moves $T$, buffer capacity $N$, exploration rate $\epsilon$
**Output:** Trained Q-networks $Q_0, Q_1$
 1: Initialize replay buffers $D_0, D_1$ with capacity $N$
 2: Initialize Q-networks $Q_0, Q_1$ with random weights $\theta_0, \theta_1$
 3: **for** $episode = 1 : M$ **do**
 4:     $x_0 \leftarrow$ empty board
 5:     **for** $t = 0 : T$ **do**
 6:         $p \leftarrow t \bmod 2$
 7:         $\hat{p} \leftarrow (p + 1) \bmod 2$
 8:         $a_t \leftarrow \begin{cases} \text{random action} & \text{with probability } \epsilon \\ \text{argmax}_a Q_p(s_t, a; \theta_p) & \text{otherwise} \end{cases}$
 9:         Execute $a_t$, observe $r_t$ and $x_{t+1}$
10:         **if** $x_{t+1}$ is terminal **then**
11:             Store $(x_t, a_t, r_t, x_{t+1})$ in $D_p$
12:             Store $(x_{t-1}, a_{t-1}, r_{t-1}, x_{t+1})$ in $D_{\hat{p}}$
13:             Update($D_p, Q_p$) using algorithm 2
14:             Update($D_{\hat{p}}, Q_{\hat{p}}$) using algorithm 2
15:             **break**
16:         **else**
17:             Store $(x_{t-1}, a_{t-1}, r_{t-1}, x_{t+1})$ in $D_{\hat{p}}$
18:             Update($D_{\hat{p}}, Q_{\hat{p}}$)
19:         **end if**
20:     **end for**
21: **end for**
___

**Algorithm 2** Update Q-network
___
**Input:** Replay buffer $D$, Q-network $Q$
**Output:** Updated Q-network $Q$
 1: Sample $(x_j, a_j, r_j, x_{j+1})$ from $D$
 2: $y_j \leftarrow \begin{cases} r_j & \text{for terminal } x_{j+1} \\ r_j + \gamma \max_{a'} Q(x_{j+1}, a'; \theta) & \text{for non-terminal } x_{j+1} \end{cases}$
 3: Gradient descent on loss $(y_j - Q(x_j, a_j; \theta))^2$
___