# OpenReview forum: "Transformers Learn Transition Dynamics when Trained to Predict Markov Decision Processes"
_EMNLP/2024/Workshop/BlackBoxNLP — BlackboxNLP 2024_

### Official Review · Reviewer_6kie · 2024-09-07

**Overall Assessment:** 3
**Confidence:** 5

**Best Paper:**

1

**Best Paper Justification:**

N/A

**Comments Questions Suggestions And Typos:**

probe strained -> probes trained

**Paper Summary:**

The paper trains deep Q-learners to play Connect Four, then trains a small GPT-style Transformer on the sequences of moves of such players, and then shows that MDP state and next action can be decoded linearly or nonlinearly with decent accuracy from the Transformer. The paper argues that this shows meaningful generalization by the Transformer, going beyond "surface statistics" in training data.

**Summary Of Strengths:**

- The experiments are properly motivated
- The experiments appear to be properly carried out
- The results are clear and presented straightforwardly.

**Summary Of Weaknesses:**

While the work seems to be solid, it does not meaningfully advance our knowledge or opinion about neural network capabilities. Essentially these same studies (albeit training networks to mimic human players, not trained MDPs as far as I know) have been done for example with Othello (cited by the authors, but there is a lot more), and indeed some knowledge of variables like game state can indeed be decoded from networks in previous work. That work triggered a number of interesting debates about the validity of using probes to determine whether a neural network "represents" something---see for example Melanie Mitchell's claims that the apparent ability to decode Othello states was in fact a property of the decoder rather than the underlying neural network. These debates are not engaged with here, nor does the study provide new information touching on those debates.

---

### Official Review · Reviewer_qch1 · 2024-09-09

**Overall Assessment:** 4
**Confidence:** 4

**Best Paper:**

1

**Best Paper Justification:**

N/A

**Comments Questions Suggestions And Typos:**

- Line 54, 59: You can use \citet for narrative citation
- Line 107: MDp $\rightarrow$ MDP
- Line 245: Remove whitespace before footnote
- Footnote 1: The full stop is missing
- Line 262: becomes $\rightarrow$ because (?)
- Section 2.3: Maybe some tokenized examples of the training data would be informative for readers
- Line 337: $\mathcal{H}_{t-1}$ for the sequence of actions
- Line 469: "probe strained" $\rightarrow$ "probes trained"

**Paper Summary:**

This paper shows that Transformers learn representations that encode information about the transition dynamics and policies of Markov Decision Processes just by being trained to predict the next state or action. This is demonstrated by training autoregressive Transformers to play Connect Four and studying their internal representations with probing classifiers.

**Summary Of Strengths:**

1) The paper is very clearly written, and the logic is easy to follow.
2) The experimental setup is simple yet effective at providing evidence for the main claim.

**Summary Of Weaknesses:**

This paper was written very clearly, so I have no glaring concerns. However, I do wonder how big an MSE of e.g. 467 is for the action prediction (or how big an MSE of 0.0272 is for the state prediction). Are there more intuitive measures that can be presented (or alternatively, are there ways to make the MSE range more intuitive to readers)?

---

### Official Review · Reviewer_t7ST · 2024-09-10

**Overall Assessment:** 3
**Confidence:** 4

**Best Paper:**

1

**Best Paper Justification:**

-

**Comments Questions Suggestions And Typos:**

-

**Paper Summary:**

The paper explores the ability of language models to abstract beyond surface-level statistics and learn underlying processes by training a transformer on ConnectFour game transcripts, with the goal of investigating whether the model can encode the parameters governing the underlying process used by agents playing the game (e.g. Q-values in the case of an agent guided by deep Q-Learning or MCTS values in the case of an agent governed by Monte-Carlo tree search). The authors present an experimental setup where the model is trained to predict the next state in the context of Markov Decision Processes (MDPs), without direct observation of transition dynamics or player policies. The paper provides evidence suggesting that the models develop emergent representations of Q-values and MCTS values by showing lower mean squared error when prompted to predict these values compared to a randomly initialized model.

**Summary Of Strengths:**

+ Well-written and clear structure: The paper is logically organized and easy to follow.

+ Clear research question: The research question is well-defined. The authors seek to investigate whether LMs, when exposed to the history of game play (in this case, ConnectFour), can encode the decision-making process behind agent actions.

+ Thorough preliminaries: The authors provide adequate background and explanation of Markov Decision Processes, Q-learning, MCTS and the game of ConnectFour, making the paper approachable even for readers unfamiliar with these concepts.

+ Reproducible experiments: The experimental design is straightforward and provides sufficient details, making it relatively easy for others to recreate the results.

**Summary Of Weaknesses:**

- MSE as a metric lacks interpretability and comparing against a random baseline is limited: While MSE is used as the primary evaluation metric, it may not be the most appropriate measure. MSE quantifies how far the model's predictions are from the ground truth but does not provide insight into whether the model captures the structure of decision-making or the policy. Also, the comparison to a random baseline is useful for establishing a lower bound, showing that the model is not making purely random predictions. However, this baseline alone is insufficient evidence to conclude that the model has learned Q-values. A random baseline merely demonstrates that the model has learned something, but it doesn’t reveal the depth of what has been learned. Additional comparisons against more meaningful baselines would provide stronger evidence of the model's ability to learn the underlying decision-making processes.
Since MSE is sensitive to outliers, focusing on predicting only the best move (the action with the highest Q-value for instance) could artificially lower the MSE. This would allow the model to achieve a lower error compared to a random baseline, even if the other Q-values for suboptimal moves are incorrect or ignored. Essentially, the model could appear to perform well simply by predicting the next optimal move, without truly capturing the entire distribution of Q-values across all possible actions. However, it's important to note that this demonstrates the model has captured some level of information, which is interesting in itself.

---

### Decision · Program_Chairs · 2024-09-19

**Decision:**

Accept

**Comment:**

Reviewers agree that the research question and experiments are clear and well-motivated. I encourage the authors to incorporate the reviewers' comments regarding the choice of metrics and extending their discussion in the camera-ready version.